# OBJECT-CENTRIC LEARNING AS NESTED OPTIMIZATION

**Michael Chang, Sergey Levine & Thomas L. Griffiths** [*]

## ABSTRACT

Various iterative algorithms have shown promising results in unsupervised decomposition simple visual scenes into representations of humans could intuitively consider objects, but all with different algorithmic and implementational design choices for making them work. In this paper, we ask what the underlying computational problem that all of these iterative approaches are solving. We show that these approaches can all be viewed as instances of algorithms for solving a particular nested optimization problem whose inner optimization is that of maximizing the ELBO with respect to a set of independently initialized parameters for each datapoint. We lastly discuss how our nested optimization formulation reveals connections to similar problems studied in other fields, enabling us to leverage tools developed in these other fields to improve our object-centric learning methods.

## 1 INTRODUCTION

How objects are represented in the mind has been a major question in philosophy (Smith, 1996) and cognitive science (Spelke, 1990) and has motivated the question of how we can build intelligent agents that learn to similarly represent what humans consider objects, without human supervision on what specific objects are (Greff et al., 2020). Works in recent years have proposed various iterative algorithms that have shown promising results in unsupervised decomposition simple visual scenes into representations of humans could intuitively consider objects (Greff et al., 2017; Van Steenkiste et al., 2018; Greff et al., 2019; Veerapaneni et al., 2020; Locatello et al., 2020; Kipf et al., 2021; Zoran et al., 2021; Singh et al., 2021). These all share a underlying theme of iteratively refining a set of independently initialized representations to decompose the observation into representations of its constituent objects, but all have different algorithmic and implementational design choices for making this work. The lack of a unifying computational framework makes it difficult to reason about why and how these methods work and how to improve them when they do not.

In this paper, we ask what the underlying computational problem that all of these iterative approaches are solving. Such a unified problem formulation could reveal connections to similar problems studied in other fields, enabling us to leverage tools developed in these other fields to improve our object-centric learning methods. Our primary contribution is a proposal for a unifying tangible problem statement under which to view the set of iterative approaches to developed for object-centric learning so far. Specifically, we show that these approaches can all be viewed as instances of algorithms for solving a particular nested optimization problem whose inner optimization is that of maximizing the ELBO with respect to a set of independently initialized parameters for each datapoint. These algorithms can be categorized as either performing meta-learned posterior inference or meta-learned parameter estimation in a factorized symmetric generative model. We show how these two categories both represent instances of the same nested optimization.

This paper contains no experiments, and we do not claim that our proposed nested optimization formulates what object-centric methods should do, but merely that it describes the problem that previously proposed methods appear to be solving. The explicit identification of this problem could provide a common language for understanding existing methods and for improving them.

In §2, we briefly introduce the class of object-centric algorithms we consider as instances of what we call **iterative amortized inference** algorithms that either **meta-learn posterior inference** or **meta-learn parameter estimation**. In §3 we show meta-learned posterior inference and meta-learned parameter estimation are both instances of the same nested optimization problem and interpret this

---

[*]`mbchang@berkeley.edu, svlevine@eecs.berkeley.edu, tomg@princeton.edu`

problem in the context of object-centric learning. The conclusion (§4) uses the nested optimization problem to discuss the connection between object-centric learning and other fields of research.

## 2 ITERATIVE AMORTIZED INFERENCE

Consider a generative model with observables $X$, local (per-observation) latents $Z$, and global (across-observations) parameters $\theta$, defining the joint distribution for a particular sample $(z, x)$ as $p(z, x; \theta) = p(z; \theta^z)p(x \mid z; \theta^{x \mid z})$. Given a datapoint $x$, the goals of statistical inference often involve estimating the parameters $\theta$ or inferring the posterior $p(z \mid x)$. Both can be achieved via variational techniques (Neal & Hinton, 1998; Dayan et al., 1995) that frame inference as maximizing the evidence lower bound (ELBO) $\mathcal{L}$ with respect to $\theta$ and an approximate posterior $q(z \mid \cdot)$:

$$\mathcal{L}(q, \theta, x) := \mathbb{E}_{z \sim q(z \mid \cdot)} [\log p(x, z; \theta) - \log q(z \mid \cdot)]. \tag{1}$$

### 2.1 CLASSICAL AND AMORTIZED INFERENCE

Classical approaches for maximizing $\mathcal{L}$ include variants of the expectation maximization (EM) algorithm (Dempster et al., 1977), which alternates between optimizing $\max_q \mathcal{L}(q, \theta, x)$ and $\max_\theta \mathcal{L}(q, \theta, x)$, using incremental (e.g., gradient descent) or analytic approaches. Given a dataset $\{x^n\}_{n=1}^N$, and assuming that $q$ can be parameterized by variational parameters $\phi$, classical iterative methods employ a fixed learning rule (Hoffman et al., 2013), for improving $\phi$ or $\theta$, e.g.

$$\phi_{t+1}^n \leftarrow \phi_t^n + \alpha \nabla_{\phi^n} \mathcal{L}(\phi_t^n, x^n), \quad \forall n \tag{2}$$

$$\theta_{t+1} \leftarrow \theta_t + \beta \sum_n \nabla_\theta \mathcal{L}(\theta_t, x^n) \tag{3}$$

which is costly to scale to high-dimensional datasets. Thus, techniques related to the variational autoencoder (Kingma & Welling, 2013; Rezende et al., 2014, VAE) amortize (Gershman & Goodman, 2014) the optimization of $\phi^n$ for each $x^n$ via an encoder that directly maps $x$ to $\phi$. However, estimating $\phi$ without iterative feedback results in poorer performance (Krishnan et al., 2018; Cremer et al., 2018) and cannot break symmetry among exchangeable unobserved variables.

### 2.2 ITERATIVE AMORTIZED INFERENCE

Several works have proposed to combine the paradigms of iterative optimization and neural networks by replacing the fixed update rule with a update network $f$ that is trained to optimize the unrolled iterative procedure (Kirsch & Schmidhuber, 2020; Andrychowicz et al., 2016) for improving ELBO. These can be categorized as performing posterior inference or parameter estimation via a meta-learning algorithm (Thrun & Pratt, 2012; Schmidhuber, 1987), with the former conducted across a single dataset (like Andrychowicz et al. (2016)) and the latter conducted across a dataset of mini-datasets (like Finn et al. (2017)).

For methods that **meta-learn posterior inference** (Marino et al., 2018b;a; 2020; Greff et al., 2019; Veerapaneni et al., 2020; Emami et al., 2021), instead of an encoder that directly maps $x^n$ to $\phi^n$, an update network $f$ improves a (initially random) previous estimate $\phi_t^n$ as $\phi_{t+1}^n \leftarrow f(\phi_t^n, \nabla_{\phi_t^n} \mathcal{L}_t)$ for each datapoint $x^n$. While $\phi^n$ is updated per-datapoint, the model parameters $\theta$ and weights of $f$ are updated across datapoints.

In contrast, methods that **meta-learn parameter estimation** (Greff et al., 2017; Van Steenkiste et al., 2018; Zoran et al., 2021; Locatello et al., 2020; Singh et al., 2021) treats each datapoint $x^n$ as *itself* a mini-dataset of $M$ measurements $x^{n,m}$ (e.g. $x^n$ is an image and $x^{n,m}$ is a pixel or feature of $x^n$). Each datapoint $x^n$ is generated from per-datapoint model parameters $\theta^n$ with per-measurement latents $z^{n,m}$, thus defining a per-datapoint ELBO $\mathcal{L}^n$. The role of the update network $f$ in this setting is to improve the (initially random) model parameters $\theta^n$ as $\theta_{t+1}^n \leftarrow f(\theta_t^n, \nabla_{\theta_t^n} \mathcal{L}_t^n)$, which generally also involves improving the per-measurement variational parameters $\phi^{n,m}$.

## 3 OBJECT-CENTRIC LEARNING AS NESTED OPTIMIZATION

We explicitly unify iterative amortized inference methods (§2) as solving a particular nested optimization problem whose inner optimization is that of maximizing the ELBO. We describe the generic

bi-level optimization problem (one-level of nesting), then show that meta-learned posterior inference and meta-learned parameter estimation instantiate this problem with one and two levels of nesting respectively. Lastly we interpret this nested optimization in the context of object-centric learning.

## 3.1 THE NESTED OPTIMIZATION PROBLEM

Consider the following bi-level optimization problem over a generic dataset $\{x^n\}_{n=1}^N$ with datapoints $x^n$. Define the parameters $\lambda^n$ as optimized *per*-datapoint, and the parameters $\mathbf{w}$ as optimized *across* datapoints. With the ELBO $\mathcal{L}$ as the inner objective and a task objective $\mathcal{J}$ as the outer objective, we express the bi-level optimization problem as

$$\min_{\mathbf{w}} \quad \sum_n \mathcal{J}\left(x^n, \mathbf{w}, \lambda_*^n\right)$$
$$\text{s.t.} \quad \lambda_*^n = \arg\max_{\lambda^n} \mathcal{L}\left(\tilde{x}^n, \lambda^n\right). \tag{4}$$

When the inner optimization is conducted via a fixed update rule, the solution of the inner problem can be embedded as a differentiable optimization layer (Amos & Kolter, 2017) within a neural network with weights $\mathbf{w}$. Here, we partition $\mathbf{w}$ as $\mathbf{w} = [\mathbf{w}_e, \mathbf{w}_d]$, where $\mathbf{w}_e$ are weights of an encoder that processes $x^n$ into $\tilde{x}^n$, and $\mathbf{w}_d$ are weights of a decoder that computes the outer objective $\mathcal{J}$ with the fixed point $\lambda_*^n$ as an input. Special cases include the case where $\mathbf{w}_e$ is the identity (i.e., no pre-processing of $x^n$) and the case where $\mathbf{w}_d$ is the identity (i.e. no post-processing of $\lambda_*^n$).

Using a trainable network $f_{\mathbf{w}}$ as the update rule instead, e.g. $\lambda_{t+1}^n \leftarrow f_{\mathbf{w}}(\lambda_t^n, x^n)$, implicitly parameterizes a constraint set $\mathcal{C}_{\mathbf{w}}(x^n)$. The weights $\mathbf{w}$ from Eq. 4 now include the weights $\mathbf{w}_u$ of the learnable update rule $f_{\mathbf{w}}$, yielding:

$$\min_{\mathbf{w}} \quad \sum_n \mathcal{J}\left(x^n, \mathbf{w}, \lambda_*^n\right)$$
$$\text{s.t.} \quad \lambda_*^n = \arg\max_{\lambda^n \in \mathcal{C}_{\mathbf{w}}(x^n)} \mathcal{L}\left(\tilde{x}^n, \lambda^n\right). \tag{5}$$

The constraint set $\mathcal{C}_{\mathbf{w}}(x^n)$ implicitly depends on $\mathbf{w}_e$, which pre-processes $x^n$, and $\mathbf{w}_u$, which updates $\lambda^n$. Any update rule that monotonically improves upon $\mathcal{L}$ is thus a fixed point operation whose fixed point locally maximizes $\mathcal{L}$ (Neal & Hinton, 1998; Wu, 1983). It is in this sense that we can understand $f_{\mathbf{w}}$ as trained to perform a fixed point operation.

## 3.2 POSTERIOR INFERENCE AND PARAMETER ESTIMATION

Now we show that iterative amortized inference for posterior inference and parameter estimation implement fixed point procedures that solve the aforementioned nested optimization problem. This is to our knowledge the first unification of both approaches under the same problem statement.

**Meta-learned posterior inference** Methods for meta-learned posterior inference (§2.2) train a VAE decoder as the generative model with parameters $\theta$ and an update network $f_{\mathbf{w}}$ that updates $\phi_{t+1}^n \leftarrow f_{\mathbf{w}}(\phi_t^n, \nabla_{\phi_t^n} \mathcal{L}_t)$ for each datapoint $x^n$. We recover the problem formulation in Eq. 5 by substituting the negative ELBO for the outer objective $\mathcal{J}$, the per-datapoint variational parameters $\phi^n$ for $\lambda^n$, and the model parameters $\theta$ for the subset of $\mathbf{w}_d$ that compute the $p(x \mid z)$ term of the negative ELBO. Then the update network implements the fixed point operation $\phi_{t+1}^n \leftarrow f_{\mathbf{w}}(\phi_t^n, x^n)$ that computes $\nabla_{\phi_t^n} \mathcal{L}_t$ from $\phi_t^n$ and $x^n$ as an initial pre-processing step.

**Meta-learned parameter estimation** Methods for meta-learned parameter estimation (§2.2) treat each datapoint $x^n$ as a mini-dataset of measurements $x^{n,m}$. We recover the problem formulation in Eq. 5 by substituting a per-datapoint ELBO $\mathcal{L}^n$ for the inner objective and let $\lambda^n := \left(\theta^n, \{\phi^{n,m}\}_{m=1}^M\right)$, meaning that the inner optimization jointly optimizes the per-datapoint model parameters $\theta^n$ and all per-measurement variational parameters $\phi^{n,m}$. Existing approaches implement this by using $f_{\mathbf{w}}$ to compute an EM (Greff et al., 2017; Van Steenkiste et al., 2018) or modified soft K-means (Locatello et al., 2020; Zoran et al., 2021; Singh et al., 2021) step. Since the variational inference problem (Eq. 1) is itself a bi-level optimization over $\theta$ and $\phi$, meta-learned parameter estimation is actually a tri-level optimization, optimizing $\mathbf{w}$ across datapoints $x^n$ at the

outer level, $\theta$ across measurements $x^{n,m}$ but per-datapoint at the middle level, and $\phi$ per-measurement at the inner level. The inner two objectives are the ELBO $\mathcal{L}^n$ defined for each datapoint and the outer objective $\mathcal{J}$ is a task objective specified for the dataset, such as image reconstruction or attribute classification (Locatello et al., 2020).

### 3.3 OBJECT REPRESENTATIONS AS INDEPENDENTLY INITIALIZED FIXED POINTS

Having established the above formalism, the object-centric learning problem as studied so far represents a subset of instances of the nested optimization problem described in §3.1, where the inner optimization is of a *set* of independently initialized parameters $\boldsymbol{\lambda}^n := \{\lambda^{n,k}\}_{k=1}^K$ that are symmetrically updated by $f_{\mathbf{w}}$. In the context of meta-learned posterior inference, the $\boldsymbol{\lambda}^n$ are variational parameters $\phi^n$ for a mixture density (Bishop, 1994) generative model, as in (Greff et al., 2019; Veerapaneni et al., 2020; Emami et al., 2021). In the context of meta-learned parameter estimation, the $\boldsymbol{\lambda}^n$ are the component parameters $\boldsymbol{\theta}^n$ of a mixture model for clustering the independent sensor measurements $x^{n,m}$ (e.g. pixels) of a datapoint $x^n$ (e.g. image), as in (Greff et al., 2017; Van Steenkiste et al., 2018; Zoran et al., 2021; Locatello et al., 2020; Singh et al., 2021). Our nested optimization unifies the entity representations $\boldsymbol{\lambda}^n$ in both contexts as the resulting fixed points of an inner ELBO optimization.

As an example, the slot attention module (Locatello et al., 2020) is a state-of-the-art object-centric learning method that performs meta-learned parameter estimation. It adds a set of empirical improvements (attention heads (Vaswani et al., 2017), layer normalization (Ba et al., 2016b), and gated recurrent unit (GRU) (Cho et al., 2014)) to a differentiable soft k-means layer that iteratively refines a set of *slots* $\lambda^{n,k}$, acting as cluster centroids, given an input $x^n$. Without these additions, slot attention is equivalent to optimizing Eq. 4 because the soft k-means algorithm is known to monotonically improve the ELBO (Bottou & Bengio, 1995). The task objectives $\mathcal{J}$ considered by (Locatello et al., 2020) included both image reconstruction and classification. In practice, Locatello et al. (2020) found that replacing the manually-defined soft k-means update with a learnable update $\boldsymbol{\lambda}^n \leftarrow f_{\mathbf{w}}(\boldsymbol{\lambda}_t^n, x^n)$ (Eq. 5) that uses attention heads, layer normalization, and a GRU improves performance.

## 4 DISCUSSION

Our nested optimization problem reveals at least four connections between the object centric learning problem and several other research directions in the field. First is the connection to the literature on fast weights (Schmidhuber, 1992; Ba et al., 2016a; Irie et al., 2021) because the parameters $\boldsymbol{\lambda}^n$ serve as the weights that are modified during the inner optimization, during execution time. Second is the connection to the literature on deep implicit layers (Duvenaud et al., 2020) which studies neural layers not as explicitly parameterized functions but as functions whose input and output satisfy some constraint, such as a fixed point procedure. Third is the connection to the literature on meta-learning (Schmidhuber, 1987; Finn et al., 2017; Thrun & Pratt, 2012; Andrychowicz et al., 2016) because the outer parameters $\mathbf{w}$ are trained to improve the inner parameters $\boldsymbol{\lambda}^n$ during execution time. Fourth is the connection to the literature on algorithmic information theory (Kolmogorov, 1965; Li et al., 2008; Solomonoff, 1964) because whereas the entity representations learned in the context of meta-learned posterior inference have an associated distribution parameterized by $\phi^{n,k}$ and thus are *statistically independent*, the entity representations $\boldsymbol{\theta}^{n,k}$ learned in the context of meta-learned parameter estimation do not have an associated distribution and thus are *algorithmically* independent samples from the distribution that randomly initialized their values (Janzing & Schölkopf, 2010, §3.2). These different fields have their own conceptual and implementation tools that could potentially improve our understanding of how to build better object-centric models and of how objects could potentially be represented in the mind.

### ACKNOWLEDGEMENTS

This work was supported by ARL, W911NF2110097, with computing support from Google Cloud Platform.

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
