# OpenReview forum: "Object-Centric Learning as Nested Optimization"
_ICLR.cc/2022/Workshop/OSC — ICLR2022 OSC  Poster_

### Official Review · Reviewer_iYd1 · 2022-03-10
**A heuristic but not pragmatic attempt**

**Rating:** 1
**Confidence:** 2

**Review:**

The authors unify the iterative algorithms in object-centric learning methods into a particular nested optimization problem with solving a maximization of ELBO. They join meta-learn posterior inference and meta-learn parameter estimation in existing methods to the same nested optimization problem and interpret it as the essence of object-centric learning. Although unification is always what scientists pursue, a simple combination is not enough. Some pivotal questions should be answered in this paper.
1. Why do we need unification in object-centric learning methods? What's the advantage of regarding these practical algorithms as a theoretical optimization formulation?
2. Is there any detailed examples that your idea can bring some difference to recent research?

---

### Official Review · Reviewer_foEG · 2022-03-16
**Well-motivated paper, needs improvement in clarity**

**Rating:** 2
**Confidence:** 2

**Review:**

The paper aims to identify the underlying computational problem that existing iterative approaches to object-centric learning are trying to solve. Specifically, the paper classifies existing approaches into two categories: those that meta-learn posterior inference and those that meta-learn parameter estimation. The paper then proposes an optimization problem that unifies these two categories, where the inner layer optimizes ELBO with respect to the per-datapoint parameters (e.g., slot representations, cluster assignments), and the outer layer optimizes the task objective (e.g., reconstruction, classification) with respect to network weights (e.g., encoder and decoder). The paper also suggests some connections to other fields.

Pros
- The paper is well-motivated. A unified problem formulation can shed light on ways to improve the existing methods.

Cons
- The clarity of the paper can be improved. For example, I didn't understand the key difference between the two proposed categories. In particular, why can't Slot Attention fit in the first category?
- I am not sure whether the proposed framework is general enough. In particular, why does the inner objective have to be ELBO? The paper mentioned that the soft k-means algorithm is known to monotonically improve the ELBO. However, in Slot Attention the soft k-means algorithm is replaced by learnable updates. It is unclear whether the learnable updates is still optimizing the same objective.

---

### Official Review · Reviewer_jmQW · 2022-03-20
**Highlights interesting connections across various approaches to object-centric representation learning**

**Rating:** 2
**Confidence:** 1

**Review:**

The authors present a unifying framework for object-centric learning, bringing together a wide array of distinct methods under a single framework. I personally find this flavor of manuscripts particularly useful, as they've previously helped me better understand fields of research (for example Cunningham and Ghahramani (2015) is in a similar spirit albeit for a different set of problems). I think the current manuscript will be a valuable addition to the workshop and serve to generate useful discussion within the community. However, one aspect of the manuscript which I felt could potentially be improved (perhaps as future iterations of the work) are the insights that can be gained from the proposed interpretation of object-centric learning. For example, given that the authors propose to interpret object-centric learning as nested optimization, perhaps there are relevant methods from the (nested) optimization literature which could now be more easily ported over and used to improve object-centric learning. Or instead, perhaps the proposed framework can be used to further outline similarities/differences between current work.

Minor/typos:
- Abstract: "promising results in unsupervised decomposition simple visual scenes .." -> "promising results in unsupervised decomposition of simple visual scenes.."



References:
- Cunningham, John P., and Zoubin Ghahramani. "Linear dimensionality reduction: Survey, insights, and generalizations." The Journal of Machine Learning Research 16.1 (2015): 2859-2900.

---

### Decision · Program_Chairs · 2022-03-23

**Decision:**

Accept (Poster)

**Comment:**

This paper is relevant to the workshop and outlines an interesting connection between iterative object-centric representation learning approaches and nested optimization. As such, I believe it can provide a valuable contribution to this workshop, despite not having shown an immediate practical advantage of this unification as pointed out by reviewer iYd1. We encourage the authors to take the reviewers' feedback into account when preparing the camera-ready version of the paper.